# Optimizing Growth of the Future Liver Remnant and Making In-Situ Liver Transsection Safe—A Standardized Approach to ISLT or ALPPS

Andrea Alexander, Nadja Lehwald-Tywuschik [ID], Alexander Rehders, Levent Dizdar, Georg Fluegen [ID], Sami Alexander Safi [#][ID] and Wolfram Trudo Knoefel *,[#]

Departments of Surgery, Heinrich-Heine-University, University Hospital Duesseldorf,
40225 Duesseldorf, Germany
* Correspondence: knoefel@hhu.de; Tel.: +49-211-8117351
# These authors contributed equally to this work.

**Abstract:** In-situ splitting of the liver before extended resection has gained broad attention. This two-step procedure requires several measures to make an effective and safe procedure. Although the procedure is performed in many institutions, there is no consensus on a uniform technique. The two steps can be divided into different parts and a standardized technique may render the procedure safer and the results will be easier to evaluate. In this paper, we describe a detailed approach to in-situ splitting that allows making both procedures safe, avoids liver necrosis, and is easily reproducible. In the first procedure the portal branches to segments I and IV to VIII are divided, the arterial branches and bile ducts to these segments are preserved and encircled and the parenchyma between segments II/III and IVa/b is divided. This avoids necrosis and bile leaks of segments I and IV and avoids urgent completion operations. In particular, the handling of vital structures close to the dissection line seems important to us. Complete splitting and securing the right and middle hepatic vein will make the second step of this procedure a minimal-risk procedure at a stage where the patient is still recovering from the more demanding first step.

**Keywords:** in-situ splitting; ISLT; ALPPS; standardized surgical technique





## 1. Introduction

Recent reports have proposed several methods to augment the future liver remnant before extended right resection of the liver [1–5]. Although portal venous embolization is well established and produces satisfactory results in many patients, the growth of segments II and III is often slow and sometimes insufficient in volume (Figures 1 and 2) [6,7]. This precludes many patients from potentially curative resection. The technique of in-situ liver transsection and portal venous division (also referred to as ISLT or ALPPS) and secondary completion hepatectomy, has gained some attention over the past decade to generate faster and more efficient growth of the liver remnant [3–5,8]. The technique of this procedure varies. Due to the high complication rate of both procedures, many modifications have been described to minimize the trauma of the first step [9–11]. However, a standardized initial operation can be performed safely and reduces the complication rate in the vulnerable phase before the second operation. To obtain optimal growth and to prevent situations that make the completion hepatectomy more urgent, the initial operation has to be designed to avoid bile leaks, venous congestion and segmental liver ischemia. For accurate planning of the two steps of this complex operation, a detailed understanding of the underlying pathology and the anatomy is of major importance. Preoperative imaging will have to identify arterial and venous blood supply and drainage as well as the biliary anatomy. We routinely use computed tomography with a portal and arterial phase. An MRCP is

reserved for additional questions regarding the biliary tree. The patient is then presented to and discussed in our multidisciplinary hepatologic tumor board before treatment.

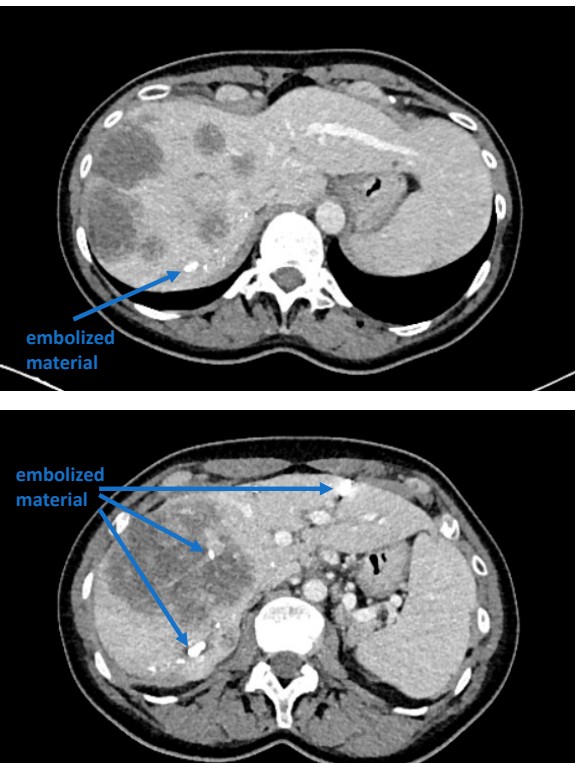

**Figure 1.** A patient with a multilocular, large hepatocellular carcinoma, a small left lobe that would not suffice to avoid liver insufficiency after resection, and insufficient portal venous embolization.

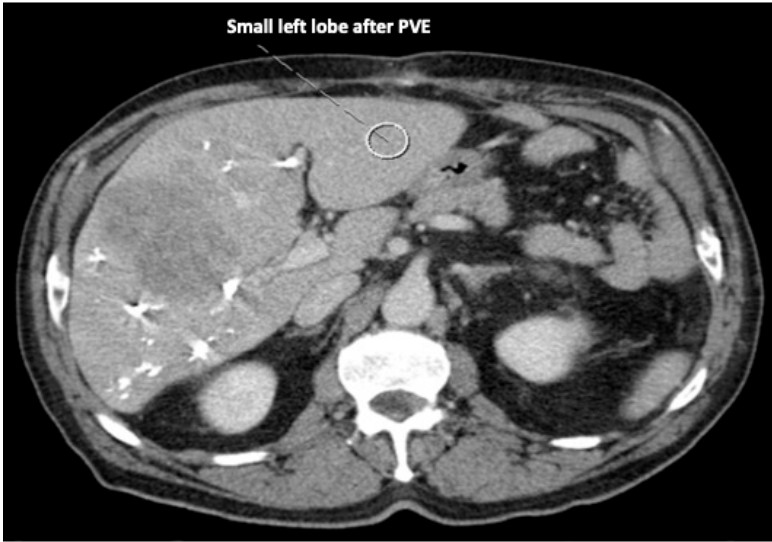

**Figure 2.** Frequently, even after careful portal venous embolization, the left lobe remains too small to enable a safe resection of the right lobe.

We describe a standardized two-step procedure that supports optimal growth, minimizes complications after the first operation and renders the second operation safe and easy.

## 2. Operative Technique

Exploration of the abdomen and liver is conducted as routinely performed for major liver resections by laparotomy. Extrahepatic spread is ruled out or confirmed to be resectable. After initial mobilization of the left and right liver, an intraoperative ultrasound is conducted to verify preoperative assessment and the resectability of the liver lesions.

For the complete mobilization of the liver off the vena cava a division of all caval branches from the renal veins up to the three hepatic veins, including all segment I branches, is then performed. The right hepatic vein is isolated next and marked with a vessel loop. To avoid injury of the middle hepatic vein, this vein is not dissected free. Instead, the left hepatic vein is encircled and one vessel loop is passed down between the right border of the left hepatic vein, then passed in front of the vena cava and then brought up along the left border of the right vein. This will encircle the middle hepatic vein. At this step, it is not important to dissect the middle hepatic vein to its adventitia. To avoid small injuries of the vein and perturbing bleeding, some connective tissue should be left on the vein.

Then, a lymphadenectomy of the hepato-duodenal ligament is conducted for oncological and/or preparatory reasons and to visualize the vascular and biliary anatomy of the hepatic hilum. A cholecystectomy is routinely performed. All right portal branches, i.e., the main right portal vein and all branches to segments I and IV of the portal vein are divided. Particular attention is attributed to preserving all arterial and biliary branches including those to the right liver. Additionally, segment I and IV branches should be preserved, if at all possible. All these branches are marked by vessel loops. This avoids liver necrosis and bile leaks that may otherwise require premature reoperation. In particular, the artery to segment IV is sometimes difficult to preserve since it crosses the left portal vein. However, complete deprivation of blood flow will cause necrosis of segment IV and eventually a bile leak.

In case a portal vein reconstruction is required for oncological reasons, e.g., a resection of the portal bifurcation, this should be performed at the initial operation at this point. Arterial reconstructions, if necessary, are equally better accomplished during the first stage of the procedure.

The biliary tree is identified, and a decision is made to preserve the extrahepatic biliary system or not. In case it should be preserved it is dissected off the hepatic tissue aiming to identify and encircle all branches that need to be divided in the future. Due to scarred tissue around the bile duct after endoscopic stenting, this may be difficult in the first procedure. In these cases, this part of the operation can be postponed to the second stage of the operation. For optimal drainage, a T-drain may be inserted at the end of the operation. In case the biliary tree needs to be resected, the left bile duct is divided between the segmental branches II and III and the branch to segment IV. The distal stump is closed, and a hepaticojejunostomy is performed to segments II and III.

To complete in-situ splitting, the liver capsule is incised on the right side of the falciforme ligament and in the sulcus arantii. The parenchyma between segments II/III and I/IV is then divided (Figure 3). The structures to be transected in the future, i.e., all arterial and biliary branches to segments I and IV-VIII, as well as the right and middle hepatic veins, are marked by vessel loops that are fastened with titanium clips avoiding strictures of the vessels (Figures 4 and 5). These remain in situ for easier identification during the second procedure. The future resection specimen is enveloped in a plastic foil (3M™ Steri ™ Drape Isolationsbeutel, 1003, 49 cm × 49 cm) to prevent adhesions in particular to the vena cava and at the resection margin.

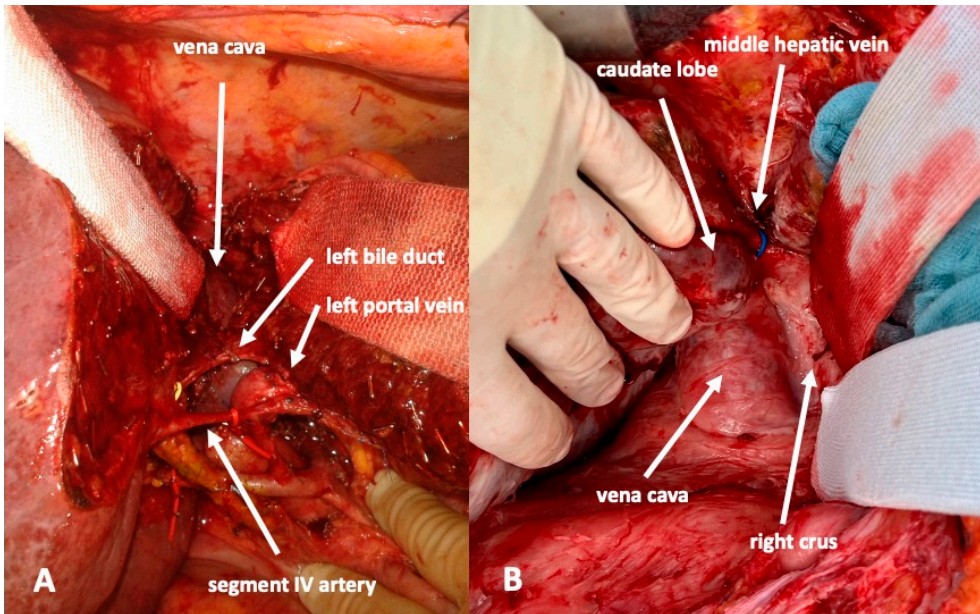

**Figure 3.** A complete transsection between liver segments II/III and I/IV is recommended to render the second procedure as easy as possible. (**A**): During the first procedure. The vena cava is easily seen. (**B**): During the second procedure after division of the arteries and bile ducts to the right and caudate lobe. Removal of the resection specimen is safe and fast without further preparation or dissection.

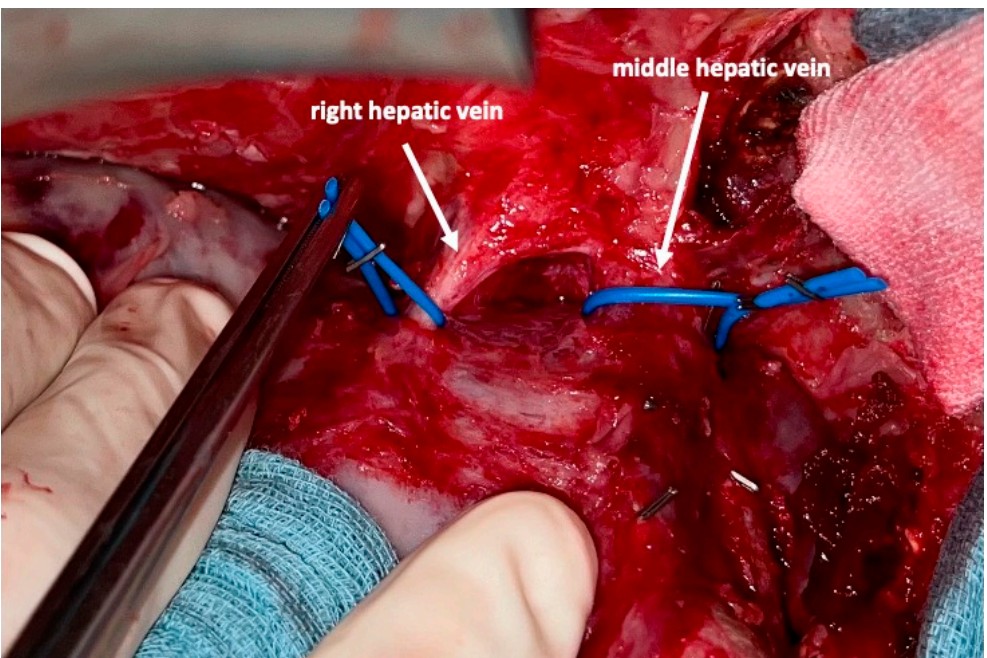

**Figure 4.** After transsection of the liver the middle and right hepatic veins are marked with a blue vessel loop each. This facilitates division of these vessels after regeneration of the left lobe.

On day 7 after the operation, CT-volumetry and, potentially, (99 m)Tc-mebrofenin-hepatobiliary-scintigraphy are performed. Alternative function tests may be used, according to the centers' experience. We strongly recommend repeating the volumetric and functional analyses with the identical technique as used before the first step of the operation. This is repeated on a weekly basis until the necessary volume and function are reached (Figure 6). After sufficient growth of the remnant liver volume, i.e., a volume of >0.6% of the body weight or >25% of functional liver volume, and a cut-off value in the

scintigraphy of 2.7%/min/m² is achieved, completion hepatectomy is scheduled. In the case of prior chemo- or immunotherapy, the cut-off value of functional liver value may vary and additional investigations such as a (99 m)Tc-mebrofenin-hepatobiliary-scintigraphy are even more important. Some authors advocate more than 35% of functional liver volume in these conditions. During the second operation, a thorough exploration of the abdomen is performed, and the plastic foil is removed (Figure 7). Under the guidance of the vessel loops the previously marked arterial and biliary branches to the resection specimen, as well as the right and middle hepatic veins are divided, and the specimen is removed. If necessary, reconstructions of vessels and biliary structures are performed (Figures 8 and 9).

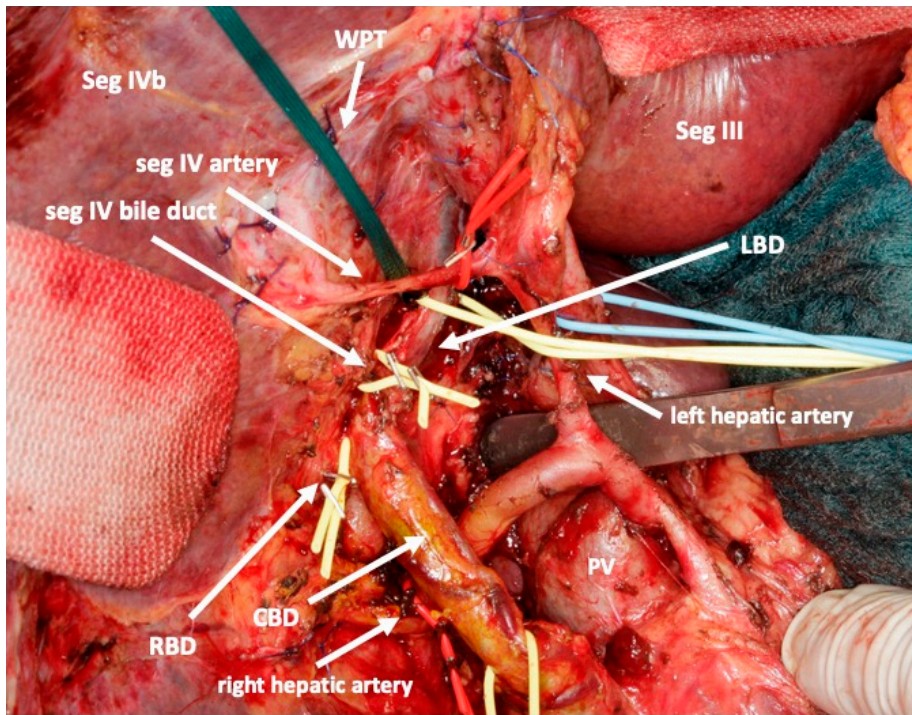

**Figure 5.** After transection of the liver and marking of the middle and left hepatic veins, all vessels to the right and caudate lobe in the hilum that need to be divided during the second operation are marked with red or yellow vessel loops. Only arteries and bile ducts are remaining. (woven polyester tape: WPT; left bile duct: LBD; right bile duct: RBD; portal vein: PV.

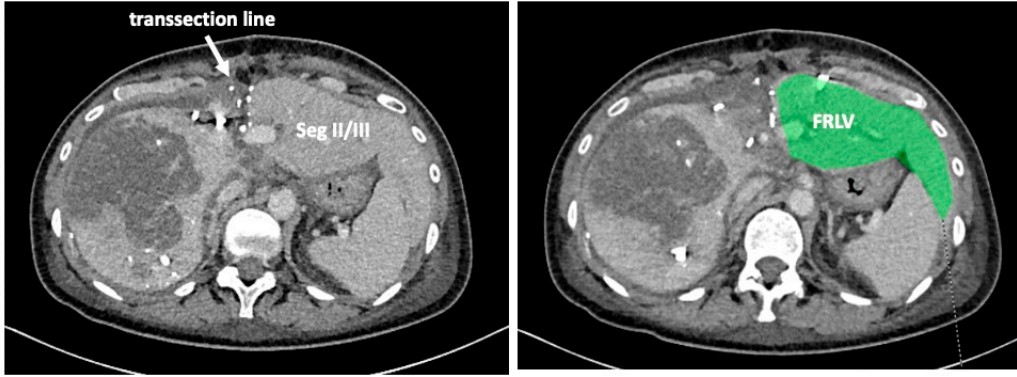

**Figure 6.** The functional residual liver volume (FRLV) is measured on a weekly basis until a sufficient volume is reached.

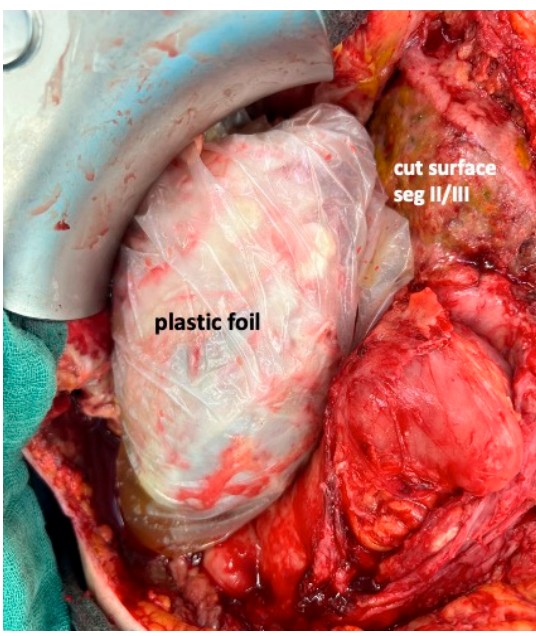

**Figure 7.** On re-exploration of the abdominal cavity, the plastic foil remains inert to the surrounding tissue and is easily removed, giving access to the remaining vasculature to the right liver lobe.

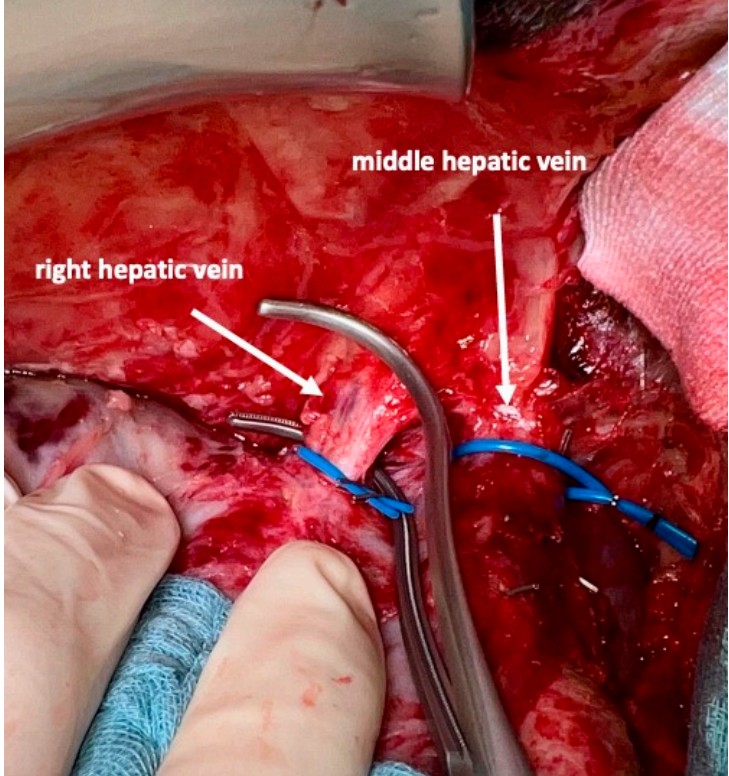

**Figure 8.** After dividing the arteries and bile ducts to the right liver lobe and segment I, the middle and right hepatic vein are very easily accessed and divided between vascular clamps. This completes the removal of segments I and IV–VIII.

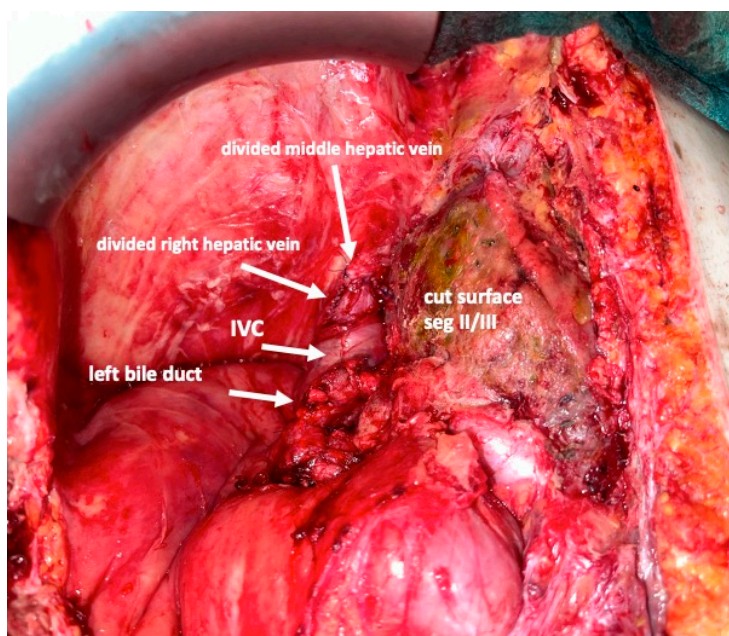

**Figure 9.** After resection is completed, only a relatively small wound surface remains and the left liver lobe is often already adherent to the surrounding structures, avoiding kinking of the left hepatic vein. If that is not the case, it is recommended to stabilize the left liver lobe by stitching Teres' ligament to the median abdominal wall.

## 3. Discussion

The technique of in-situ liver transsection described above is safe, and reproducible and induces growth of segments II and III at an excellent rate. Morbidity and mortality were not increased significantly in comparison to portal venous embolization with this method in recent studies [12]. The functional liver remnant regeneration rate of 63 to 75% after 4 to 9 days [4,5] compares favorably with that after portal venous embolization of 10 to 62% after 1 to 60 weeks [5,7,13–15].

The particular technique described herein induces regeneration significantly faster than portal venous embolization and also offers potential as a salvage procedure in case of insufficient growth after portal venous embolization [5,13].

The functional liver remnant growth of ISLT or ALPPS has repeatedly been shown to be significantly faster and/or better than after portal venous embolization alone in our series as well as in others [5,7,13]. It seemed difficult to compare these data to other ALPPS or ISLT techniques because the technical details were not as explicitly mentioned in previous papers. In addition, we have established and repeatedly used this technique as a salvage procedure after portal venous embolization. In a case series from our institution, portal venous embolization, when successful, produced a mean functional liver remnant volume to body weight ratio growth from 0.49% +/− 0.17% to 0.67% +/− 0.05% whereas ISLT as a salvage procedure produced a growth from 0.42% +/− 0.08% to 0.81% +/− 0.09%. In 12 of 13 cases, this produced significant growth sufficient to complete the second step [13].

The particularity of the described technique lies in an anatomically oriented total portal flow deprivation of segments I and IV-VIII, including the complete division of segments II and III from the right lobe and segment I. It also emphasizes the effort to preserve the arterial blood supply to segments I and IV as well as the biliary drainage. Recent opinions and comparisons of non-standardized techniques favored a less invasive approach in the first step, leaving much of the dissection, eventual reconstruction and some of the parenchymal division to the second step. However, the regenerative stimulus will be greater after total portal dissection to the right side and, since regeneration occurs at a relatively fast pace, patients undergo the second step earlier than after portal venous

embolization. They are more vulnerable to liver insufficiency during and after the second step because more functional liver volume has been removed and physical impairment is usually more significant at this stage. This prompted us to establish a standardized approach to this formidable procedure after we had seen a mortality rate in the first procedures that called for improvement.

After adopting the technique described herein in 2017 ($n = 25$) we saw an improvement in our mortality rate from 30.7% to 12%. This compares favorably to the mortality rate in the recent report on early adopting centers when considering the fact, that a majority of our patients were relatively old ($n = 16 > 70$ years old), suffered from cholangiocellular carcinoma ($n = 13$), underwent simultaneous venous ($n = 7$) or arterial reconstructions ($n = 4$), and/or underwent salvage procedures after portal venous embolization ($n = 7$) [9].

Several variants of the technique of in situ splitting of the liver are used at different centers. Since no consensus has been reached so far on using a uniform technique, we will discuss the advantages of the procedure we established in a standardized fashion at our institution.

### 3.1. Arterial Blood Supply

The dissection of the artery to segment IV off the left portal vein in the recessus of Rex can be difficult and time consuming. In addition, the parenchymal dissection and the preparation of the left biliary tree is much easier after transsection of this artery since the space between segments II/III and segments IV/I opens up after this transsection. Other groups, therefore, sacrifice the arterial blood supply to segment IV [4]. Some of these groups also avoid dissection of the bile duct to segment IV. While this may enhance liver growth of segments II and III, we consider the risk of segment IV necrosis a risk for complications between steps 1 and 2. The preservation of arterial blood flow to segments IV, as described in this report, reduces the risk of liver necrosis in this segment. Liver necrosis predisposes to abscesses and biliary leaks. Both impair the regeneration of the liver and may force the surgeon to intervene prematurely. While the preservation of all arteries at the first step prevents complications during the interval before the second stage, the dissection of the arterial anatomy, especially the identification of accessory arteries and the bifurcation into left and right hepatic arteries prevents injuries to these structures at the second stage of the operation. This detailed dissection of the arterial branches requires time and meticulous dissection especially of arteries to segments I and IV. This can be challenging in cases where the tumor is very large or previous operations have altered the tissue quality. However, during the second step, this enables a fairly rapid procedure. Sometimes, scarring may obscure the view of these important structures after a prolonged waiting time. This risk is minimized by dissection and marking with vessel loops at a moment where tissue alteration is at its minimum. Arterial involvement and the necessity to perform an arterial reconstruction are not as frequent as portal vein reconstruction. In case the bifurcation of the common hepatic artery is involved, it is safer to perform this part in the second step. This situation is, however, very rare in our experience. Most frequently the left hepatic artery is involved at the base of segment IVb and this may be in combination with portal venous infiltration. In order to have free access to the recessus of Rex and to the ventral aspect of segment I, we find it more convenient to perform this resection in the first procedure. It may, however, result in partial ischemia of segments I and IV in case these arteries are involved as well. Then, a delay of arterial resection needs to be discussed. Like after portal venous reconstruction, it is imperative to avoid pressure on the liver during abdominal closure. We had to combine four arterial reconstructions with this technique in our recent series. One patient developed a thrombotic complication possibly due to a previously undiagnosed heparin-induced thrombocytopenia. We frequently opt for leaving the abdominal wall open temporarily until no major volume alterations are expected.

### 3.2. Biliary Tree

It is of major importance to avoid the occurrence of a bile leak before the scheduled second operation since this can lead to septic complications with a delayed regeneration of the future liver remnant. It may even require a premature completion operation with all risks associated with a resection before the designated volume and function are reached [16]. The dissection of the extrahepatic biliary tree can be difficult in cases of a larger tumor burden in segment IV. Frequently, the left bile duct shares an intimate contact to segment IVb and the identification of the bile duct to segment I can be challenging. In our experience, it is, however, safer to explore the biliary anatomy at this stage than to risk a bile leak. Occasionally, the biliary anatomy at the base of segment IV, when dissected, predisposes to an injury and a bile leak. Therefore, in these cases, a dissection of the biliary tree or the biliary reconstruction can be postponed and can be conducted during the second procedure. In case the extrahepatic bile duct is too close to the tumor, shows an impaired blood supply, or sustained critical injuries during the dissection, it is safest to perform the final reconstruction of the biliary tree at the first stage. This may include a hepaticojejunostomy to the bile ducts to segments II/III. Again, this minimizes the risks of a bile leak and secondary interventions.

### 3.3. Portal Venous Blood Supply

To obtain optimal growth, it is of importance to divide as many portal venous branches as possible into the segments that need to be resected, usually segments I, IV, V, VI, VII, and VIII. Complete division of portal branches requires transsection and ligation or oversewing and not only ligation. This may lead to revascularization during a prolonged waiting time [17].

To enable the division of all portal branches, the main portal vein and the left branch to segments II and III need to be mobilized of the right liver and segments IV and I, including all dorsal branches. This results in a completely mobile portal vein that is only connected to the left lateral segments. This mobilization may be difficult because of tumor growth approaching or involving the area of the portal bifurcation or the proximal left portal vein. If that is the case, portal vascular resection and reconstruction need to be conducted at this stage to avoid insufficient interruption of portal flow to the right liver segments. A delay of liver regeneration due to a complication has to be avoided and portal venous reconstructions may cause complications. Portal resection is usually necessary at the portal bifurcation. We find it very difficult to leave the portal vein fixed to the right liver and still take down all branches to Segments I and IV to VIII. Reconstruction may be sometimes easier without the right specimen in situ but can be achieved safely in our experience and is important to obtain a maximum regenerative stimulus by dividing all branches to the right specimen. Using this technique of in-situ liver transsection we had to perform seven portal venous reconstructions in our recent series. One patient developed a thrombosis of the left portal vein that was successfully treated by local removal and lysis by ileocolic venous access. Another patient developed a thrombus of the main portal vein in conjunction with a heparin-induced thrombocytopenia and a thrombotic occlusion of the left artery. An attempt to remove the thrombotic material surgically was futile. As mentioned for arterial reconstructions, we prefer to leave the abdominal wall open for the first postoperative period to avoid pressure on the liver.

### 3.4. Parenchymal Transsection

To optimize regeneration, complete parenchymal transsection is also important because intraparenchymal collaterals contribute to portal blood flow [18]. Although 'partial ALPPS' may be easier for the first stage, this time is then invested in the second stage and the risk of a delayed second procedure may be increased. Once the liver is completely mobilized off the cava with the division of all direct hepatic branches, it is usually not very demanding to encircle the left hepatic vein, pass a 5 mm woven polyester tape in the sulcus of arantii and from there behind the portal vein, the left arteries and the left bile duct

between segments II/III and segments I/IV. Once this guide is established, transsection can be performed under slight tension on this tape that secures all vital structures at the same time.

*3.5. Hepatic Veins*

All direct venous drainage of the liver into the inferior vena cava is divided. All main hepatic veins are preserved at the first stage of the procedure. However, we emphasize the need to address all three veins individually and encircle them to facilitate the second stage of this operation. Additionally, handling of the liver, control of eventual bleeding and parenchymal transsection is much safer once this is conducted. During the second stage, simple placement of vascular clamps on the middle and right hepatic vein suffices to remove the specimen after the division of the corresponding feeding arteries and draining biliary branches.

## 4. Conclusions

In case the growth of the liver or recuperation from the initial operation is slower than anticipated, it is even more important that nothing forces the surgeon to reoperate. This may well be the case in cirrhotic or fibrotic livers or in patients that have undergone chemotherapy [19]. A standardized approach as described above prepares the patient for optimal growth, minimizes the risk of complications and does not force the surgeon into the second stage of the procedure.

Although this technique has been standardized for extended right resections, analogous operations can be performed for any liver resection, e.g., left hepatectomies, on the occasion that the future liver remnant volume is marginal [3,20]. This may be of particular help in patients with poor liver quality or fibrosis.

The second operation is technically much less demanding once the initial stage has been performed as described and causes no major trauma. This is of importance in patients with significant comorbidities in particular. The discussion concerning the amount of surgery that is conducted in the first step of this procedure is quite vivid. Our motivation to complete the most difficult steps in the first procedure originates from the perception, that patients are usually in a better condition before the first procedure. As long as the liver to be resected is not removed, liver insufficiency is very unlikely to occur, regardless of the extent of the dissection. In contrast, patients before the second procedure are still recovering from major surgery, often somewhat frail and their liver function may be marginal at times or simply difficult to estimate correctly. These factors prompted us to perform the crucial steps in the first procedure. In order to make results comparable, it is suggested to report the technique used for in-situ liver transsection in detail in every study.

**Author Contributions:** Conceptualization, A.A. and W.T.K.; methodology, A.A. and S.A.S.; validation, N.L.-T. and A.R.; formal analysis, L.D. and G.F.; investigation, A.A., A.R. and W.T.K.; resources, A.R. and S.A.S.; data curation, N.L.-T., L.D. and G.F.; writing—original draft preparation, A.A.; writing—review and editing, S.A.S. and W.T.K.; visualization, A.A., S.A.S. and L.D.; supervision, W.T.K.; project administration, A.A. and W.T.K. All authors have read and agreed to the published version of the manuscript.

**Funding:** This research received no external funding.

**Institutional Review Board Statement:** This study was approved by the local institutional review board (Heinrich-Heine-University, Duesseldorf, Germany; study-no.: 2018-258-KFogU). All procedures performed in this study were in accordance with the ethical standards in the 1964 Declaration of Helsinki and its later amendments.

**Informed Consent Statement:** Informed consent was obtained from all subjects involved in the study.

**Data Availability Statement:** The data presented in this study are available on request from the corresponding author.

**Conflicts of Interest:** The authors declare no conflict of interest.

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
