# Peer review of "Optimizing Growth of the Future Liver Remnant and Making In-Situ Liver Transsection Safe—A Standardized Approach to ISLT or ALPPS"

_curroncol, doi:10.3390/curroncol30030249_

Round 1
Reviewer 1 Report
The authors present the study entitled " Optimizing growth of the future liver remnant and making in-2 situ liver transection safe. A standardized approach to ISLT or ALPPS.". The authors present an high quality standardized procedure but some issues need to be clarified. The current trend is to be as minimally invasive as possible and the technique presented here departs from the current recommendations in ALPPS.
- The authors suggested in the manuscript a complete mobilization of the liver off the vena cava a division of all caval branches from the renal veins up to the three hepatic veins, including all segment I branches, is then performed. The right hepatic vein is isolated next and marked with a vessel loop. According to our experience a complete mobilization should be avoid due to the higher number of adherences produced by this maneuver which increase the difficulty of the second stage.
- The authors recommend portal vein or arterial reconstruction in the first stage but there is consensus agreement of the experts in ALPPS about try to be less invasive as possible in the first stage to avoid complications which can delay liver regeneration and the second stage.
- The authors recommend >25 % of functional liver volume, but surgical indication of ALPPS is related to CRLM with neoadjuvant chemotherapy and at least >35 % of functional liver volume should be recommended.
- Plastic foil it is not recommended in ALPPS and the authors should not use it.
Author Response
Thank you for your critical appraisal of our paper.
The authors present the study entitled " Optimizing growth of the future liver remnant and making in-2 situ liver transection safe. A standardized approach to ISLT or ALPPS.". The authors present an high quality standardized procedure but some issues need to be clarified. The current trend is to be as minimally invasive as possible and the technique presented here departs from the current recommendations in ALPPS.
The discussion concerning the amount of surgery that is done in the first step of this procedure is, as you mentioned, quite vivid. Our motivation to complete the most difficult steps in the first procedure originates from our perception, that patients are usually in a better condition before the first procedure. As long as the liver to be resected is not removed, liver insufficiency is very unlikely to occur, regardless of the extent of the dissection. In contrast, patients before the second procedure are still recovering from major surgery and their liver function may be marginal at times or simply difficult to estimate correctly. These factors prompt us to perform the crucial steps in the first procedure. This position statement has been added to the discussion.
- The authors suggested in the manuscript a complete mobilization of the liver off the vena cava a division of all caval branches from the renal veins up to the three hepatic veins, including all segment I branches, is then performed. The right hepatic vein is isolated next and marked with a vessel loop. According to our experience a complete mobilization should be avoid due to the higher number of adherences produced by this maneuver which increase the difficulty of the second stage.
Your concern regarding the adhesions between vena cava and liver is very justified and stems from your experience with reparative liver surgery. To circumvent the potential risks involved, we wrap the entire right specimen in a plastic foil that can be removed with ease during the second procedure and preserves all critical dissection planes. This, more detailed explanation was added to the 'operatic technique' section.
- The authors recommend portal vein or arterial reconstruction in the first stage but there is consensus agreement of the experts in ALPPS about try to be less invasive as possible in the first stage to avoid complications which can delay liver regeneration and the second stage.
The timing of vascular resection and reconstruction certainly is worth to be discussed. We agree, that a delay of liver regeneration due to a complication has to be avoided, as you mentioned. Portal resection is usually necessary at the portal bifurcation. We find it very difficult to leave the portal vein fixed to the right liver and still take down all branches to Segments I and IV to VIII. Reconstruction may be sometimes easier without the right specimen in situ but can be achieved safely in our experience and is important to obtain a maximum of regenerative stimulus. Arterial involvement is not as frequent and needs a separate discussion. In case the bifurcation of the common hepatic artery is involved, it is safer to perform this part in the second step. This situation is, however, very rare in our experience. Most frequently the left hepatic artery is involved at the base of segment IVb and tis may be in combination with portal venous infiltration. In order to have free access to the recessus of Rex and to the ventral aspect of segment I we find it more convenient to perform this resection in the first procedure. It may, however, result in partial ischemia of segments I and IV in case these arteries are involved as well. Then, a delay of arterial reaction needs to be discussed. We have added these thoughts to the discussion.
- The authors recommend >25 % of functional liver volume, but surgical indication of ALPPS is related to CRLM with neoadjuvant chemotherapy and at least >35 % of functional liver volume should be recommended.
Your point concerning neoadjuvant therapy, in particular for CRLM, is very valuable and needed to be added. We find it difficult to set a standard in this condition and favor further functional assessment. We have added this important point in the 'operative technique' section.
- Plastic foil it is not recommended in ALPPS and the authors should not use it.
We are aware of some discussion on the use of plastic foils in these conditions. However, we have used this foil since we first described the procedure (3) and have not encountered any problems so far. The foil is completely inert and designed to cover the intestines in-situ (3M™ Steri ™ Drape Isolationsbeutel, 49 cm x 49 cm). Without the foil, the second procedure becomes a lot more demanding as you mentioned already when discussing adhesions. We have, consequently, retained our suggestion in the manuscript.
Thank you once again for your insightful comments.
Reviewer 2 Report
In this manuscript (curroncol-2191878), the authors describe their technique for performing a two stage extended right hepatectomy (meaning resection of S1, S4, S5, S6, S7, S8) with in situ liver splitting, including a first step operation consisting in the closure of portal vein to all segments to be resected, while guaranteeing integrity of arterial flow to and of biliary drainage from such segments. Such first step also includes the transection of the liver parenchyma between the left later sector and S4, on the right side of the falciform ligament insertion. In addition, a wide mobilization of the right liver is performed during such first step, with ligation of eventual S1 veins and accessory hepatic veins, as well as a wide dissection of the three hepatic veins. The second step, which will be performed when the future remnant liver volume will be considered sufficient for performing a safe liver resection, will consist in the transection of S1, S4 to S8 biliary and arterial branch, of the right and middle hepatic veins, and in the removal of the devascularized liver. This is basically a “How I do it” manuscript, where the authors describe their own technique. I have some comments: In the title and introduction, while introducing their surgical technique, the authors refer to ALPPS and ISLT. However, it is not clear how their technique is different from the two above mentioned. The authors should expand on peculiarities of their technique. A peculiarity of the technique herein described seems to me the dissection and ligation of p4, while preserving the artery and bile duct for S4. Such strategy is related to the risk of S4 postoperative necrosis (with related risk of bile leak) in case of transection of S4 arterial and bile ducts. However, the dissection of p4(a and b) from S4 arterial and bile ducts may be challenging and difficult to perform. I recommend the authors to clearly explain their approach to the dissection of such structures. In addition, while the total closure of G4, when simultaneous to liver transection, may increase the risk of S4 necrosis, some evidence suggested that S4 vascular deprivation may enhance FLR volume increase. I suggest the authors to expand on this topic. The use during the first step operation of a plastic bag to envelop the liver to be resected and of vessel loops to encircle arteries and bile ducts and hepatic veins which will be transected during the second step operation is quite questionable: should the patient never undergo a second step operation, foreign material may cause inflammatory reaction or abdominal infection: please comment. Line 58: it is not clear how encircling the left hepatic vein may reduce the risk for middle hepatic vein injury. Instead, I believe that the dissection of the middle and left hepatic vein should be avoided during the first step operation, if not strictly necessary, in order to reduce the risk of facing with adhesions at these levels during the second step operation, otherwise avoidable. It is not clear why eventual arteriale, vein and biliary reconstruction are recommende during the first step procedure: please explain. Line 136: The authors state that their technique is safe, reproducible and induces S2-3 growth at an excellent rate. However, in the complete absence of data regarding their experience, this seems to me an overstatement. I suggest the authors to report the data regarding their experience in order to legitimate their conclusions.
Author Response
We thank this reviewer for the critical remarks and the suggestions to improve our manuscript.
-In the title and introduction, while introducing their surgical technique, the authors refer to ALPPS and ISLT. However, it is not clear how their technique is different from the two above mentioned. The authors should expand on peculiarities of their technique. A peculiarity of the technique herein described seems to me the dissection and ligation of p4, while preserving the artery and bile duct for S4. Such strategy is related to the risk of S4 postoperative necrosis (with related risk of bile leak) in case of transection of S4 arterial and bile ducts. However, the dissection of p4(a and b) from S4 arterial and bile ducts may be challenging and difficult to perform. I recommend the authors to clearly explain their approach to the dissection of such structures.
ISLT or ALPPS is a strategy that optimizes liver regeneration of the FLR and that, and the general surgical approach, is not at all different from the technique described. As you stated, we wanted to highlight some peculiarities that render the procedure safer and less prone to complications between step I and II. We also agree, that the minute dissection of the portal branches p4 and p1 may be challenging, especially when the arterial and biliary branches need to be preserved. Some clarifying remarks were added to the manuscript. We consider these steps crucial to avoid complications and to gain time, if needed, between the two steps.
-In addition, while the total closure of G4, when simultaneous to liver transection, may increase the risk of S4 necrosis, some evidence suggested that S4 vascular deprivation may enhance FLR volume increase. I suggest the authors to expand on this topic.
We agree with your comment as in fact this has been suggested in the literature. However, in our experience, the risk of necrosis and bile leaks outweighs the small increase in growth factors resulting from total closure of G4. Our main focus is to reduce complication rate – hence our concern about G4 closure. We added a comment in the text.
-The use during the first step operation of a plastic bag to envelop the liver to be resected and of vessel loops to encircle arteries and bile ducts and hepatic veins which will be transected during the second step operation is quite questionable: should the patient never undergo a second step operation, foreign material may cause inflammatory reaction or abdominal infection: please comment.
We are aware of some discussion on the use of plastic foils and vessel loops in these conditions. However, we have used this foil since we first described the procedure (3) and have not encountered any problems so far. The foil is completely inert and designed to cover the intestines in-situ (3M™ Steri ™ Drape Isolationsbeutel, 49 cm x 49 cm). Without the foil and the vessel loops, the second procedure becomes a lot more demanding. In the very rare case, where the patient never undergoes the second step, we have removed all foreign material in a small second procedure. This occurred once in our series since 2009. We have, consequently, retained our suggestion in the manuscript.
Line 58: it is not clear how encircling the left hepatic vein may reduce the risk for middle hepatic vein injury. Instead, I believe that the dissection of the middle and left hepatic vein should be avoided during the first step operation, if not strictly necessary, in order to reduce the risk of facing with adhesions at these levels during the second step operation, otherwise avoidable.
This comment is crucial and our wording was certainly misleading. The point was to highlight, that it is safest to avoid working on the middle hepatic vein. We consider it safe to encircle the right and left veins, unless there is a common trunk for middle and left vein. The vessel loop around the middle vein is really encircling all tissue between the vena cava, the right border of the left vein and the left border of the right vein. This area can then very easily be identified for transection in the second step. This point has been better explained in the ‘Operative Technique’ chapter.
It is not clear why eventual arteriale, vein and biliary reconstruction are recommende during the first step procedure: please explain.
This point was also raised by reviewer 1. We may copy our answer:
The timing of vascular resection and reconstruction certainly is worth to be discussed. We agree, that a delay of liver regeneration due to a complication has to be avoided, as you mentioned. Portal resection is usually necessary at the portal bifurcation. We find it very difficult to leave the portal vein fixed to the right liver and still take down all branches to Segments I and IV to VIII. Reconstruction may be sometimes easier without the right specimen in situ but can be achieved safely in our experience and is important to obtain a maximum of regenerative stimulus.
Arterial involvement is not as frequent and needs a separate discussion. In case the bifurcation of the common hepatic artery is involved, it is safer to perform this part in the second step. This situation is, however, very rare in our experience. Most frequently the left hepatic artery is involved at the base of segment IVb and this may be in combination with portal venous infiltration. In order to have free access to the recessus of Rex and to the ventral aspect of segment I we find it more convenient to perform this resection in the first procedure. It may, however, result in partial ischemia of segments I and IV in case these arteries are involved as well. Then, a delay of arterial reaction needs to be discussed. We have added these thoughts to the discussion.
We recommended biliary reconstruction in the first step in case the biliary tree may show an impaired blood supply, is infiltrated by the tumor or has sustained injuries during the procedure. This is all done to avoid a bile leak. We understand your point, however, that a biliary reconstruction can be safely done and is easier to perform in the second procedure. The timing, therefore, has to be selected individually. We have added a comment in the discussion on the biliary tree.
Line 136: The authors state that their technique is safe, reproducible and induces S2-3 growth at an excellent rate. However, in the complete absence of data regarding their experience, this seems to me an overstatement. I suggest the authors to report the data regarding their experience in order to legitimate their conclusions.
These data are obviously of importance. We did not include this in the paper since we focused on the surgical approach. Our data are included in several papers (5,9,12,13). After adopting the described technique in 2017 (n=25) we saw an improvement in our mortality rate from 30.7% to 12%%. This compares favorably to the mortality rate in the report on early adopting centers when considering the fact, that a majority of our patients were relatively old (n=16 >70 yo), suffered from cholangiocellular carcinoma (n=13) and/or were salvage procedures (n=7). We have added a paragraph on this topic in the discussion.
The FLR growth of ISLT or ALPPS has repeatedly shown to be significantly faster and/or better than after PVE alone in our series as well as in others (5,7,13). In a case series from our institution, PVE, when successful, produced a growth from 0.49 +/- 0.17% to 0.67 +/-0.05% whereas ISLT as a salvage procedure produced a growth from 0.42 +/- 0.08% to 0.81% +/- 0.09%. It seemed difficult to compare these data to other ALPPS or ISLT techniques because the technical details were not as explicitly mentioned in previous papers. In addition, we have established and repeatedly used this technique as a salvage procedure after PVE. In 12 of y13 cases this produced significant growth sufficient to complete the second step (13). This is now also mentioned in the discussion.
We thank this reviewer for the many helpful comments.
Reviewer 3 Report
The manuscript entitled “Optimizing growth of the future liver remnant and making in-situ liver transsection safe. A standardized approach to ISLT or ALPPS.” by Alexander and coworkers describes the technique of the Duesseldorf group to perform an ALPPS procedure. The manuscript is well written and contains well arranged figures.
However, some points might be addressed and are listed below:
- The authors describe the preparation of the hilum and show very nice pictures regarding this issue. Since hilar structures, particularly the bile duct, tend to be quite variable in a significant number of patients, detailed preoperative planning is necessary. Therefore, a section on the preoperative imaging modalities that are usually performed/recommended before ALPPS should be added.
- Similar, is there any preoperative liver function test performed before going for an ALPPS procedure?
- The authors mention the (99m)Tc-mebrofenin-hepatobiliary-scintigraphy before the second step of the procedure. Is this also used before the 1st step to assess the function of the liver remnant? What is the cut-off set by the authors to go for the 2nd step?
- The authors state that a lymphadenectomy of the hepato-duodenal ligament is performed due to oncological reasons. What about ALPPS procedure in patients with colorectal liver metastases of hepatocellular carcinoma? In these patients, lymphadenectomy is usually not necessary.
- The authors propose to encircle the middle hepatic vein. Is a direct closure of the vein also performed during the 1st step? This might lead to even more pronounced growth of the remnant.
- Figure 5 shows a well dissected hilum with vessel loops particularly on the biliary structures. Do the authors really dissect the hilum like this in every patient? I suggest that the perfusion of the bile duct might be impaired even in case a resection of the extrahepatic bile duct is not necessary.
- Figure 7 shows the envelopment of the specimen with plastic foil to avoid adhesions. To me this is critical since even upon very careful dissection and transection, bile leakages may occur. This may lead to infections that significantly impair liver regeneration but also bring the patient into a septic condition. Moreover, in some patients even ALPPS does not lead to sufficient hypertrophy. What is the strategy in these patients regarding the plastic foil?
Author Response
We thank this reviewer for his comments and suggestions and hope that our additions to the manuscript will help the reader to better appreciate our technique.
- The authors describe the preparation of the hilum and show very nice pictures regarding this issue. Since hilar structures, particularly the bile duct, tend to be quite variable in a significant number of patients, detailed preoperative planning is necessary. Therefore, a section on the preoperative imaging modalities that are usually performed/recommended before ALPPS should be added.
We have added a paragraph to the introduction to address this issue. We agree, that the anatomy may vary considerably. However, as you certainly agree, subtle dissection of the hilum reveals the anatomy sufficiently. This dissection is always part of step 1 at our institution. We also may refer to issue #4 of your review.
- Similar, is there any preoperative liver function test performed before going for an ALPPS procedure?
Yes, we routinely perform a (99m)Tc-mebrofenin-hepatobiliary-scintigraphy before the first step as well. We have added a comment in lines 110 to 114. However, we have not discussed different techniques of volumetric and functional assessment since this may overburden our manuscript.
- The authors mention the (99m)Tc-mebrofenin-hepatobiliary-scintigraphy before the second step of the procedure. Is this also used before the 1st step to assess the function of the liver remnant? What is the cut-off set by the authors to go for the 2nd step?
Our cut-off value used for adults is 2.7%/min/m2 (de Graaf et al 2010 PMID 19937195 and PMID 20080899). We have added this value in line 117.
- The authors state that a lymphadenectomy of the hepato-duodenal ligament is performed due to oncological reasons. What about ALPPS procedure in patients with colorectal liver metastases of hepatocellular carcinoma? In these patients, lymphadenectomy is usually not necessary.
We agree that lymphadenectomy is not mandatory for some indications that may require an in- situ split procedure. We therefore have modified this sentence and included ‘and/or preparatory’ reasons. These preparatory reasons are based on the necessity to fully understand the anatomy for example to be able to preserve the arterial supply to segment 4.
- The authors propose to encircle the middle hepatic vein. Is a direct closure of the vein also performed during the 1st step? This might lead to even more pronounced growth of the remnant.
We are aware of this intriguing approach to further enhance liver regeneration. We are, however, somewhat concerned about congestion, that may cause venous oozing, necrosis and potential bile leaks. We have not yet gained any experience with closure of the middle hepatic vein in this procedure.
- Figure 5 shows a well dissected hilum with vessel loops particularly on the biliary structures. Do the authors really dissect the hilum like this in every patient? I suggest that the perfusion of the bile duct might be impaired even in case a resection of the extrahepatic bile duct is not necessary.
The concern about vitality of the bile during extensive dissections of the hilum is very reasonable. Our goal is to identify the biliary anatomy and to address all bile ducts to avoid a leak that may force us to intervene prematurely. This advantage outweighs, in our view, the risk of a necrosis of the bile duct.
- Figure 7 shows the envelopment of the specimen with plastic foil to avoid adhesions. To me this is critical since even upon very careful dissection and transection, bile leakages may occur. This may lead to infections that significantly impair liver regeneration but also bring the patient into a septic condition. Moreover, in some patients even ALPPS does not lead to sufficient hypertrophy. What is the strategy in these patients regarding the plastic foil?
The use of a plastic foil is a point of discussion. We use a foil that is designed to be left in-situ for a longer period of time (line 100). Our efforts during step1, and our emphasis in this report, are focused on avoiding septic situations and bile leaks. In case they do occur, the patient has to be treated accordingly, often by re-do surgery, to control the problem. In these rare cases, the foil is removed. We have not experienced a septic situation where the foil maintained the infection but the focus could have been controlled without an operation.
In case the patient never qualifies for completion of the resection, we remove the foil in a limited operation. Since this is relatively unlikely to occur, the small risk of another, otherwise unnecessary, operation seems to be legitimate.
Thank you once again for the many suggestions.
Round 2
Reviewer 1 Report
We thank the authors for making the proposed changes. The quality of the manuscript has improved significantly with the added comments.
Reviewer 2 Report
The authors adequately answered to my comments and recommendations.
I think the manuscript has been sufficiently improved for eventual publication in Current Oncology.